# Patient-level predictors of diabetes-related lower extremity amputations at a quaternary hospital in South Africa

Sifiso Mtshali[ID]*[◐], Ozayr Mahomed[◐]

Discipline of Public Health Medicine, School of Nursing and Public Health Medicine, University of KwaZulu Natal, Durban, South Africa

◐ These authors contributed equally to this work.
* smtshali@wol.co.za

## Abstract

### Introduction

Diabetes-related lower extremity amputation has a major psycho-social and economic cost on the patient as well as a direct impact on financial expenditure within health facilities.

### Aim

This study aimed to determine the incidence and patient-related factors related to diabetes-related amputations amongst patients that were referred to the quaternary hospital between 1 January 2014 and 31 December 2015.

### Methods

A retrospective cohort study. Data were retrieved from the medical record for each diabetes patient that was managed at IALCH during the study period. The following variables were collected: sociodemographic parameters (age, gender, and ethnicity) and diabetes-related parameters (type of diabetes) and additional complications.

### Results

Ninety-nine patients (0, 73%) of all diabetes patients managed were new diabetes-related lower-extremity amputations. There were statistically significant increased odds of female patients (OR: 1, 7) and patients with non-insulin dependent diabetes (OR: 1, 64) to have new diabetes-related amputations. Patients older than 60 years (OR: 1, 31); African patients (OR: 1, 35) patients with cardiovascular complications (OR: 1, 04) and patients with retinopathy (OR: 1, 48) were more likely to have diabetes-related amputations but not statistically significant.

### Conclusions

A combination of primary preventive strategies, early detection and appropriate management of patients with diabetes and specific guidelines on the frequency, clinical and

**Data Availability Statement:** All relevant data are within the paper and its Supporting Information files.

**Funding:** The authors received no specific funding for this work.

**Competing interests:** The authors have declared that no competing interests exist.

laboratory tests required for early diagnosis and referrals with early signs of diabetes-related complicationsat primary care level will assist in reducing the long term adverse outcomes including amputations.

## 1. Introduction

Diabetes mellitus is an important contributor to morbidity and mortality globally. In 2019, approximately 463 million people (20–79 years) were living with diabetes globally, and the figure is expected to increase to more than 700 million by 2045 [1]. Seventy-nine percent of people with diabetes live in low-income and middle-income countries, with 77% of deaths due to diabetes in Africa occurring in individuals younger than 60 years of age [1]. The costs associated with the management of diabetes and its related complication were estimated to account for 12% of world healthcare expenditure (US$673 billion) in 2015 and is expected to increase to over US$802 billion by 2040 [2].

Mortality from diabetes-related complications in South Africa ranked amongst the top ten causes of death and increased from 5.1% in 2014 to 5.5% in 2016 [3]. An estimated 1 in 8 people or 4,5 million adults ($\geq 20$ to $\leq 79$ years) were living with diabetes in 2019 in South Africa [1]. The ever-increasing burden of diabetes in South Africa is associated with rapid demographic and socio-economic changes, coupled with an ageing population, rapid urbanisation and a sedentary lifestyle [4].

The impact of the rapid growth of the diabetes burden on health services is made worse by the high proportion of patients with poorly controlled diabetes. Findings from the first South African national survey on non-communicable diseases, the South African National Health and Nutrition Examination Survey (SANHANES-1 (2011–2012), showed that only half (51%) of patients diagnosed with and receiving treatment for diabetes had controlled blood glucose (HbA1c< 7%) [5]. Other studies conducted in South Africa showed that between 11.2–20% of diabetes patients had controlled blood glucose [6].

Chronic hyperglycaemia is associated with long-term complications caused by damage, dysfunction, and different organ failure. Hyperglycaemia induces tissue damage through mitochondrial superoxide production [7]. The major long-term complications of diabetes are macro-vascular (peripheral vascular disease (PVD), cerebrovascular accident (CVA), and coronary artery disease (CAD); and microvascular (retinopathy, neuropathy, and nephropathy) in nature [8].

Lower extremity amputation (LEA) is not always a medical complication of diabetes as with coronary heart disease, including myocardial infarction (MI), nephropathy, or retinopathy in which the respective organ failure is directly associated with diabetes, but occurs as a result of disease progression [9]. PVD and neuropathy lead to a loss of sensation in the lower extremities and subsequently lead to LEA [10]. The progressive peripheral neuropathy leads to loss of sensation, which may additionally lead to trauma, proprioception challenges and wasting of muscles. These pathologies negatively affect the weight-bearing areas under the foot leading to ulceration, which becomes prone to infections. The impaired blood supply to the skin and failure of the biomechanics of the foot, neuro-sensory loss eventually lead to amputation [11].

In the United States of America, the incidence of diabetes-related non-traumatic LEA has shown an increase of 50% between 2009 and 2015 from the 43% decline between 2000 and 2009 [12]. Similarly, in the United Kingdom, there was an increase of 19, 4% of diabetes-related LEAs between 2014 and 2017 compared to 2010 to 2013 [13]. A tertiary-level care

based retrospective cohort study conducted in the diabetes clinic of Komfo Anokye Teaching Hospital, Ghana showed that the average incidence of diabetes-related LEA increased from 0.6% (95% CI: 0.21–2.21) in 2010 to 2.4% (95% CI: 1.84–5.61) per 1000 follow up years in 2015 [14]. Data obtained from the provincial health information system of KwaZulu-Natal in South Africa for the five years, 2013 to 2017, indicated that there was an increase in the rates of diabetes-related LEA in the hospitals across the province [15].

In addition to the significant psycho-social and socio-economic cost diabetes-related LEA have on the patient and their families, the repeated contact of the patient with the health care system and the long-term management have a direct negative financial impact. The academic hospital is the apex institution within the province and ideally sees patients referred from lower-level facilities. The patients referred to the central hospital usually have multiple co-morbidities and are high-risk patients. Although several studies were conducted on factors associated with diabetes-related LEA, very few have addressed this specific group of patients. The aim of the study was to determine the overall incidence of LEA (above and below knee) and associated risk factors for LEA amongst diabetes patients that were referred to the Central hospital (IALCH) between 1 January 2014 and 31 December 2015.

## 2. Materials and methods

### 2.1 Study design and setting

This was an retrospective cohort study of all diabetes patients that were consulted at Inkosi Albert Luthuli Central Hospital (IALCH), KwaZulu-Natal, South Africa, for the period of 1 January 2014 to 31 December 2015 (2 years). IALCH is an 864 bedded quaternary hospital providing specialist and sub-specialist level of health care in Durban, and serves as a referral hospital for the province of KwaZulu-Natal and the Eastern parts of the Eastern Cape.

### 2.2 Participants

All adult ($\geq$18 years) diabetes patients who were treated at IALCH during the study period were included in the study.

### 2.3 Data collection

IALCH has an electronic health record system that allows data storage and retrieval. A business intelligence system functions at the back-end of the system. In order to collect the required data, the initial step involved retrieving the database with all patients with diabetes based on ICD 10 codes- E 10.1 and E14.9. Thereafter, we inserted the amputation ICD 10 code and CPT procedure code to identify all diabetes patients with above and below knee new and previous LEA. The following data (age, gender, race, type of diabetes, medical compliactions) were retrieved for the patients an exported to Microsoft Excel.—- The data were exported from Microsoft Excel into STATA version 13 software for analysis. The following variables were retrieved from the electronic database: Sociodemograhic variables including Age, gender, race, type of diabetes, hospital status, additional diabetes related medical complications and lower extremity amputation (new or present).

### 2.4 Statistical analysis

Descriptive statistics, in the form of frequencies and proportions for categorical data and measures of central tendency, were used for continuous data. Bivariate and multivariate logistic regression were utilised to determine the predictive variables.

## 2.5 Ethics and permissions

The Biomedical Research Ethics Committee (BREC) of the University of KwaZulu-Natal, Durban granted the ethical approval for the study (Protocol reference number: BREC REF: BE221/17). Gatekeeper permission was acquired from IALCH and the Provincial Health Research and Knowledge Management. Data were retrieved from an electronic database, with no patient level identifiers. The ethics committee waivered requirements for Informed consent.

## 3. Results

During the period under investigation (January 2014 to December 2015) there were 13742 patients with diabetes attending IALCH representing 3.5% of all outpatient consultations (400 495). We used a total sample of 13495 as 277 patients were mothers with diabetes in pregnancy. Over these two years, 234 (17 per 1000) of the diabetes patient attending IALCH had an ICD_ 10 code related to amputations associated with their diagnosis. Ninety-nine (7 per 1000) of the diabetes patients attending IALCH were coded as new amputations conducted at IALCH, accounting for 2% of all diabetes admissions.

Ninety percent (89) of new diabetes-related LEA was more than 50 years of age, with a mean age of 60, 95 (SD: 10, 49) a median age of 59 years (IQR: 53–70 years). Fifty-seven percent (56) of new diabetes-related LEA were female, 54% (53) were of South African Indian, 37% (36) were African, and 77% (76) were non-insulin dependent diabetic patients (Table 1). The mean length of stay (LOS) for new diabetes-related LEA was 11,65 days (SD: 14,41) with a median LOS of 6 days (IQR: 2–14 days).

## 3.1 Patient-level determinants

New diabetes-related LEA and all patients with amputations shared several risk factors with some variations since the new patients were part of the total population. Bivariate analysis indicated that there were a statistically significant increased odds of females (OR: 1,76); Black African patients (OR: 1,39) and patients with cardiovascular complications (OR: 1, 05) to have new diabetes-related LEA. Patients older than 60 years (OR: 1,25); patients with non-insulin

**Table 1. Demographic profile of all patients with amputation at IALCH.**

| Variables | All Amputations | | Pre-existing amputation | | New Amputation | |
|---|---|---|---|---|---|---|
| | Number | Percentage | Number | Percentage | Number | Percentage |
| **Age (years)** | | | | | | |
| 18–24 | 4 | 2% | 4 | 3% | 0 | 0% |
| 25–49 | 30 | 13% | 20 | 15% | 10 | 10% |
| >50 | 200 | 85% | 111 | 82% | 89 | 90% |
| **Gender** | | | | | | |
| Female | 139 | 59% | 83 | 61% | 56 | 57% |
| Male | 95 | 41% | 52 | 39% | 43 | 43% |
| **Race** | | | | | | |
| SA Indian | 128 | 56% | 75 | 58% | 53 | 54% |
| African | 81 | 36% | 45 | 35% | 36 | 37% |
| Mixed Race | 12 | 5% | 8 | 5% | 4 | 4% |
| White | 6 | 3% | 2 | 1% | 4 | 4% |
| Other | 1 | 0% | 0 | 0% | 1 | 1% |
| **Diabetes Type** | | | | | | |
| Insulin dependent | 69 | 29% | 46 | 34% | 23 | 23% |
| Non-insulin dependent | 165 | 71% | 89 | 66% | 76 | 77% |

**Table 2. Bivariate and multivariate analysis of demographic determinants of diabetes-related LEA.**

|  | New Amputations | | | | | |
| Variables | Unadjusted OR | 95% CI | p value | Adjusted OR | 95% CI | p value |
| --- | --- | --- | --- | --- | --- | --- |
| Age >60 | 1,25 | 0,81–1,93 | 0,23 | 1,31 | 0,86–2,00 | 0,21 |
| Females | 1,76 | 1,12–2,84 | 0,04* | 1,7 | 1,09–2,64 | 0,02* |
| Black African | 1,39 | 0,90–2,09 | 0,04* | 1,35 | 0,87–1,95 | 0,2 |
| Non-insulin dependent diabetes | 1,6 | 0,99–2,68 | 0,82 | 1,64 | 1,01–2,65 | 0,04* |
| Medical complications | 1,07 | 0,70–1,70 | 0,53 | 1 | 0,45–2,2 | 0,99 |
| Cardiovascular | 1,05 | 0,69–1,60 | 0,02* | 1,04 | 0,53–2,07 | 0,9 |
| Nephropathy | 0,84 | 0,30–1,92 | 0,4 | 0,88 | 0,34–2,24 | 0,79 |
| Neuropathy | 0,63 | 0,13–1,92 | 0,23 | 0,58 | 0,16–2,07 | 0,4 |
| Retinopathy | 1,24 | 0,69–2,13 | 0,85 | 1,48 | 0,72–1,97 | 0,29 |
| PVD | 0,93 | 0,47–1,69 | 0,84 | 0,99 | 0,50–1,97 | 0,99 |

*p<0.05 **p<0.001

dependent diabetes (OR: 1,6) and patients with retinopathy (OR: 1,24) were more likely but not statistically significant to have new diabetes-related LEA (Table 2). After multivariate analysis, there were a statistically significant increased odds of female patients (OR: 1, 7) and patients with non-insulin dependent diabetes (OR: 1, 64) to have new diabetes-related LEA. Patients older than 60 years (OR: 1, 31); Black African patients (OR: 1, 35) patients with cardiovascular complications (OR: 1, 04) and patients with retinopathy (OR: 1,48) were more likely but not statistically significantly to have new diabetes-related LEA (Table 2).

## 4. Discussion

Two hundred and thirty-four (1, 73%) out of all diabetes patients attending the hospital between January 2014 and December 2015 had had an ICD_10 code related to amputations associated with their diagnosis. Ninety-nine patients (0,73%) of all diabetes patients attending IALCH were coded as new amputations conducted at IALCH, accounting for 2% of all diabetes admissions. Sixty-seven percent (66) of the new diabetes-related LEA patients had one or more medical complications indicating a need for specialised services. The above findings show a much lower rate of diabetes-related LEA than the 3% prevalence reported from three tertiary hospitals in Ghana [16] and the overall rate of major amputation of 11.1% reported from a retrospective review of medical records of consecutive type 2 diabetes patients referred for diabetes management to King Abdullah University Hospital (KAUH) in the period between January 2014 and December 2015 [17]. Several different studies have reported rates of 2%–16% [18].

South Africa, although facing human resource and financial constraints within the public health sector, has a hierarchical hospital system starting from the district hospital that offers generalised services and may offer uncomplicated LEA, depending on the available skills. Patients requiring more advanced or specialised services may be referred to the regional or tertiary hospital, where specialist services are available. IALCH, a quaternary hospital, is at the apex of the pyramid receiving referrals for patients requiring specialised services. Data analysed from the District Health Information indicated that the majority of diabetes-related LEAs were performed at Regional (level 2) and Tertiary hospitals (level 3) [19] thereby reducing the burden on the quaternary service.

Numerous factors were identified that correlate with higher amputation risk, and most were in keeping with previous studies. Other studies have indicated that male gender [20], old

age [21] and more prolonged duration with diabetes are independent predictors of diabetes-related LEA. Our study found that there is the statistically significant association between patients with non-insulin dependent diabetes (OR: 1, 71) and LEA. Our explanation is that patients in our study had diabetes for a prolonged duration, as demonstrated by the mean age of diabetes-related LEA patients of 60,95 years (SD: 10, 25), a median age of 60 years (IQR: 53–70 years), as well as possible poor glycaemic control based on their referral to a quaternary service. Blood glucose control tends to deteriorate with time after the diagnosis. As a result, this poor control leads to long term effects of hyperglycaemia with a longer duration with diabetes [22]. However, our study found a statistically significant increased odds of diabetes-related LEA amongst females. A similar finding of an increased odds of diabetes-related LEA amongst females was found in a matched case-control study conducted in the Southwest of Iran [23].

Persistent hyperglycaemia increases the risk of microvascular and macrovascular complications of diabetes, which are regarded as risk factors for diabetes-related LEA [24]. In our study, diabetes-related retinopathy was a contributing factor towards LEA, although not statistically significant. This finding is in contrast to the those of a clinic-based case-control study in North-eastern Australia's Townsville Hospital conducted between 1 January 2011, and 31 December 2013, that identified that retinopathy was not only as a contributing factor but was a most significant factor leading to amputation [25].

Diabetes patients with ischaemic heart disease [26] and hypertension [27] showed an increased risk of LEA. In our study, we grouped patients with cardiovascular diseases and CVA into a single category. We found that patients with cardiovascular complications were at a slightly increased but not significant risk of diabetes-related LEA.

In several publications, the presence of PVD was cited as a contributing factor to diabetes-related LEA [21]; however, in our study, we were unable to demonstrate any increased risk associated with PVD.

## 4.1 Study limitations

Some limitations were identified for the study. Firstly, missing data were inevitable because our analysis was a retrospective study and was dependent on the clinician completing all fields within the electronic medical record system. Several determinants such as smoking, alcohol, and lipid profile were not retrieved. Secondly, our study was conducted at a quaternary hospital that serves as a referral hospital receiving patients with multiple co-morbidities. The type of patient and nature of complications may not be representative of all patients with diabetes-related LEA. This analysis, despite having limitations for a developing country with limited data on LEA, could be justified by the fact that guidelines for the appropriate management of patients with high risk factors for diabetes-related LEA could be implemented (as this is a highly specialised institution) and the studies risk factors could potentially be modified during clinical practices. Information related to the patients diabetic control (Hb1AC levels) were not collected and poor diabetes control could have been a significant contributor to adverse outcomes including amputations. In addition, the lack of data on HIV status and antiretroviral treatment status in a high HIV burden country limits the ability to determine the role of HIV as a predictor for diabetes related LEA.

## 5. Conclusion and recommendation

In our study of patients with diabetes admitted to a quaternary care facility in South Africa patients with non-insulin dependent diabetes and who were female are at a statistically significantly increased risk for diabetes-related LEA, with older age (> 60 years), cardiovascular co-morbidities and diabetic retinopathy being non-significant contributing factors. Although not

all diabetes-related LEAs can be prevented, a combination of primary preventive strategies addressing unhealthy lifestyle factors, early diagnosis and appropriate management of diabetes at primary health care level will go a long way in reducing the complications. Specific guidelines should be developed for primary health care practitioners on the frequency, clinical and laboratory tests required for early diagnosis of diabetes-related complication and referrals to prevent long term adverse outcomes including amputations. s. We suggest that prospective studies and multicentre designs involving more detailed study should be undertaken for further conclusion as there are a paucity of studies conducted in South Africa on risk factors associated with diabetes-related LEA, especially in a country with a high burden of HIV and AIDS.

## Supporting information

**S1 Data.**
(XLSX)

## Author Contributions

**Conceptualization:** Sifiso Mtshali.

**Data curation:** Sifiso Mtshali, Ozayr Mahomed.

**Formal analysis:** Sifiso Mtshali.

**Investigation:** Sifiso Mtshali.

**Methodology:** Sifiso Mtshali, Ozayr Mahomed.

**Project administration:** Sifiso Mtshali.

**Resources:** Sifiso Mtshali.

**Software:** Ozayr Mahomed.

**Supervision:** Ozayr Mahomed.

**Validation:** Ozayr Mahomed.

**Visualization:** Sifiso Mtshali.

**Writing – original draft:** Sifiso Mtshali.

**Writing – review & editing:** Sifiso Mtshali, Ozayr Mahomed.

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
