## [Decision Letter · Decision Letter 0]

7 Aug 2020

PONE-D-20-15807

Patient-level predictors of diabetes-related lower extremity amputations at a Quaternary Hospital in South Africa

PLOS ONE

Dear Dr. Mtshali,

Thank you for submitting your manuscript to PLOS ONE. After careful consideration, we feel that it has merit but does not fully meet PLOS ONE’s publication criteria as it currently stands. Therefore, we invite you to submit a revised version of the manuscript that addresses the points raised during the review process.

We look forward to receiving your revised manuscript.

Kind regards,

Manal S. Fawzy, Ph.D., M.D.

Academic Editor

PLOS ONE

Journal Requirements:

2. In the ethics statement in the manuscript and in the online submission form, please provide additional information about the patient records used in your retrospective study. Specifically, please ensure that you have discussed whether all data were fully anonymized before you accessed them and/or whether the IRB or ethics committee waived the requirement for informed consent. If patients provided informed written consent to have data from their medical records used in research, please include this information.

3. Please address the following:

- Please describe your study variables in further detail within your Methods section.

- Please state the date(s) upon which these medical records were accessed.

Additional Editor Comments (if provided):

According to the reviewer's evaluation, the manuscript will need extensive work to meet PLOS ONE criteria of publication. The authors should pay attention to cover every concern raised by the reviewers.

Reviewers' comments:

Reviewer's Responses to Questions

**Comments to the Author**

1. Is the manuscript technically sound, and do the data support the conclusions?

Reviewer #1: Yes

Reviewer #2: No

2. Has the statistical analysis been performed appropriately and rigorously? 

Reviewer #1: I Don't Know

Reviewer #2: No

3. Have the authors made all data underlying the findings in their manuscript fully available?

Reviewer #1: Yes

Reviewer #2: Yes

4. Is the manuscript presented in an intelligible fashion and written in standard English?

Reviewer #1: Yes

Reviewer #2: Yes

5. Review Comments to the Author

Reviewer #1: PLoS ONE review – PONE-D-20-15807

This retrospective cohort study aimed to determine the incidence of patient related factors associated with diabetes related amputation among patients referred to the quaternary care hospital in KwaZulu-Natal, South Africa.

This paper has some strengths – the use of the entire diabetic population hospitalized at the quaternary site permits a reasonable estimate of amputation as a complication among those hospitalized with diabetes. There is a paucity of information about diabetes outcomes in sub-Saharan Africa, and thus this is a positive addition to the literature. The comments are mainly around increasing clarity of study definitions and data sources.

Comments:

1. The introduction starts by saying “Diabetes account for most morbidity and mortality globally”. I am not sure that is really accurate – if it is – then a reference to the statement would be important. Its certainly an important contributor, but not sure if it accounts for most morbidity and mortality.

2. The objective at one point says “incidence” and at one point says “prevalence” for lower extremity amputation – these are not interchangeable.

3. The definition of the cohort with diabetes should be better described in the methods. The data collection section say that data were retrieved “for each diabetes patient that was managed during the study period”. Does this mean people with a known diagnosis of diabetes at admission? People who met a certain blood glucose or HbA1c criterion?

4. How was data on the outcome of lower extremity amputation obtained from the electronic record – was this done by billing code for the surgery? By discharge diagnosis? A better definition of the study outcome would be helpful. Similarly, how was “type of diabetes” ascertained and defined?

5. The authors note that 99 of 234 amputations were new amputations conducted at the quaternary care facility. How were amputations performed elsewhere ascertained?

6. It is confusing that the paper seems to be all about hospitalized patients, but there is a row for “never admitted” in the table. Is it that people weren’t admitted for their amputation? This needs to be clarified.

7. Were all amputations or only those classified as “new” included in the models?

8. A major limitation is lack of measures of diabetes control (HbA1c) as a predictor. This should be mentioned in the limitations section.

9. In the conclusion, would temper the statement in the first sentence with – “In our study of patients with diabetes admitted to a quaternary care facility in South Africa, patients with NIDDM and who were female…..”

10. The authors mention HIV in the second to last sentence. The interaction between HIV and diabetes is a complex one. Without information on HIV status and ART regimen, I am not sure anything can be concluded about that interaction from this paper, although I do agree that this interaction is important. Perhaps they can make the lack of data on HIV status and ART a study limitation, which would make this concluding statement about HIV have some additional context.

Reviewer #2: In their manuscript “Patient-level predictors of diabetes-related lower extremity amputations at a quaternary hospital in South Africa”, the authors describe 234 patients who underwent amputation among 13,742 patients with diabetes.

1. Were these toe amputations? Above-knee amputations? Below-knee amputations? Did any patient have more than one amputation? These data points need to be determined.

2. It is unclear what the difference is in Table 2 between the crude analyses and the adjusted analyses.

3. The discussion initially describes a rate of 234 amputations out of 13742 patients (1.70%) as 1.73%, and then for some reason changes the numbers to 99 out of the same denominator. Next, several other regions are quoted for context, but it is not clear to the reader why these regions are chosen. Overall this is confusing.

4. The conclusions state that better screening is needed, but this is incidence data. What does that have to do with the need for better screening?

6. PLOS authors have the option to publish the peer review history of their article (what does this mean?). If published, this will include your full peer review and any attached files.

Reviewer #1: No

Reviewer #2: No

<gdiv></gdiv>

---

## [Author Response · Author response to Decision Letter 0]

19 Aug 2020

All the concerns have been addressed. Thank you for your feedback.

---

## [Decision Letter · Decision Letter 1]

16 Sep 2020

PONE-D-20-15807R1

Patient-level predictors of diabetes-related lower extremity amputations at a Quaternary Hospital in South Africa

PLOS ONE

Dear Dr. Mtshali,

Thank you for submitting your manuscript to PLOS ONE. After careful consideration, we feel that it has merit but does not fully meet PLOS ONE’s publication criteria as it currently stands. Therefore, we invite you to submit a revised version of the manuscript that addresses the points raised during the review process.

We look forward to receiving your revised manuscript.

Kind regards,

Manal S. Fawzy, Ph.D., M.D.

Academic Editor

PLOS ONE

Reviewers' comments:

Reviewer's Responses to Questions

**Comments to the Author**

1. If the authors have adequately addressed your comments raised in a previous round of review and you feel that this manuscript is now acceptable for publication, you may indicate that here to bypass the “Comments to the Author” section, enter your conflict of interest statement in the “Confidential to Editor” section, and submit your "Accept" recommendation.

Reviewer #1: (No Response)

2. Is the manuscript technically sound, and do the data support the conclusions?

Reviewer #1: Yes

3. Has the statistical analysis been performed appropriately and rigorously? 

Reviewer #1: I Don't Know

4. Have the authors made all data underlying the findings in their manuscript fully available?

Reviewer #1: Yes

5. Is the manuscript presented in an intelligible fashion and written in standard English?

Reviewer #1: Yes

6. Review Comments to the Author

Reviewer #1: 1. I found the first sentence of the abstract confusing, and suggest deleting it.

2. The authors note that the purpose of the study is to estimate prevalence, but the title contains with word “incidence” – again these are not interchangeable.

3. The authors note in response to R2 that all amputations were included (above and below knee) – this should be specified in the manuscript as well.

4. Some of the new text in Data Collection needs a bit of editing/spell checking, but appreciate the addition of ICD10 and CPT codes to the methods.

7. PLOS authors have the option to publish the peer review history of their article (what does this mean?). If published, this will include your full peer review and any attached files.

Reviewer #1: No

---

## [Author Response · Author response to Decision Letter 1]

23 Sep 2020

We have addressed all the concerns. thank you for your feedback.

---

## [Decision Letter · Decision Letter 2]

30 Sep 2020

Patient-level predictors of diabetes-related lower extremity amputations at a Quaternary Hospital in South Africa

PONE-D-20-15807R2

Dear Dr. Mtshali,

We’re pleased to inform you that your manuscript has been judged scientifically suitable for publication and will be formally accepted for publication once it meets all outstanding technical requirements.

Kind regards,

Manal S. Fawzy, Ph.D., M.D.

Academic Editor

PLOS ONE

Reviewers' comments:

Reviewer's Responses to Questions

**Comments to the Author**

1. If the authors have adequately addressed your comments raised in a previous round of review and you feel that this manuscript is now acceptable for publication, you may indicate that here to bypass the “Comments to the Author” section, enter your conflict of interest statement in the “Confidential to Editor” section, and submit your "Accept" recommendation.

Reviewer #1: All comments have been addressed

Reviewer #3: All comments have been addressed

2. Is the manuscript technically sound, and do the data support the conclusions?

Reviewer #1: Yes

Reviewer #3: Yes

3. Has the statistical analysis been performed appropriately and rigorously? 

Reviewer #1: I Don't Know

Reviewer #3: Yes

4. Have the authors made all data underlying the findings in their manuscript fully available?

Reviewer #1: Yes

Reviewer #3: Yes

5. Is the manuscript presented in an intelligible fashion and written in standard English?

Reviewer #1: Yes

Reviewer #3: Yes

6. Review Comments to the Author

Reviewer #1: The authors have generally been responsive to the comments. In the discussion – they repeatedly talk about diabetic retinopathy as a factor leading to lower extremity amputation. This is probably not what is meant – it is a factor associated with lower extremity amputation, because its is a marker of vasculopathy, but doesn’t itself lead to the amputation. That part of the discussion should be reworded for clarity.

Reviewer #3: The authors have adequately addressed the concerns raised by the reviewer.

Page 12: change ICD_10 to ICD-10

Table in page 3: need to remove blank rows.

Table page 14: remove extra columns

7. PLOS authors have the option to publish the peer review history of their article (what does this mean?). If published, this will include your full peer review and any attached files.

Reviewer #1: No

Reviewer #3: **Yes: **Eman Toraih

<gdiv></gdiv>

---

## [Editor Report · Acceptance letter]

9 Oct 2020

PONE-D-20-15807R2 

Patient-level predictors of diabetes-related lower extremity amputations at a Quaternary Hospital in South Africa 

Dear Dr. Mtshali:

I'm pleased to inform you that your manuscript has been deemed suitable for publication in PLOS ONE. Congratulations! Your manuscript is now with our production department. 

Kind regards, 

on behalf of

Professor Manal S. Fawzy 

Academic Editor

PLOS ONE